# Light absorption enhancement of black carbon in a pyrocumulonimbus cloud

Payton Beeler [1,2], Joshin Kumar[1], Joshua P. Schwarz[3], Kouji Adachi [4], Laura Fierce [2], Anne E. Perring[5,6], J. M. Katich[3,6,7] & Rajan K. Chakrabarty [1] ✉

Pyrocumulonimbus (pyroCb) firestorm systems have been shown to inject significant amounts of black carbon (BC) to the stratosphere with a residence time of several months. Injected BC warms the local stratospheric air, consequently perturbing transport and hence spatial distributions of ozone and water vapor. A distinguishing feature of BC-containing particles residing within pyroCb smoke is their thick surface coatings made of condensed organic matter. When coated with non-refractory materials, BC's absorption is enhanced, yet the absorption enhancement factor ($E_{abs}$) for pyroCb BC is not well constrained. Here, we perform particle-scale measurements of BC mass, morphology, and coating thickness from inside a pyroCb cloud and quantify $E_{abs}$ using an established particle-resolved BC optics model. We find that the population-averaged $E_{abs}$ for BC asymptotes to 2.0 with increasing coating thickness. This value denotes the upper limit of $E_{abs}$ for thickly coated BC in the atmosphere. Our results provide observationally constrained parameterizations of BC absorption for improved radiative transfer calculations of pyroCb events.

Large-scale wildfires are expected to increase in frequency due to climatic changes such as warmer springs, longer dry seasons, drier soils, and drier vegetation[1]. In some cases, wildfires can produce pyrocumulonimbus (pyroCb) clouds, a subset of cumulonimbus clouds which form from heat driven convection stemming from wildfires[1–3]. The formation of a pyroCb cloud is often referred to as a pyroCb event. In their most extreme form, pyroCb events can inject wildfire smoke into the upper troposphere and lower stratosphere (UTLS), where it can remain suspended for long periods of time and affect stratospheric temperatures and composition[2,4,5]. One of the most important constituents of pyroCb-injected smoke is black carbon (BC), a strong absorber of incoming light and one of the most potent short-lived climate warmers[6,7].

BC makes up around 2% of the total particle mass in pyroCb smoke, yet the contribution of BC to UTLS light absorption and associated warming is significant[5]. Yu et al.[2] reported BC from an August 2017 pyroCb event over western Canada to have warmed the local UTLS air by up to 7 K[2]. Correlated with this warming, large spatial anomalies in ozone and water vapor concentrations were observed corresponding to the plume location in the UTLS. Owing to its long atmospheric lifetime, BC from pyroCb events not only enhances shortwave radiation absorption, but also impacts the spatial distributions of UTLS atmospheric trace species. Tang et al.[8] reported that a single episode of pyroCb wildfire in Australia alone accounted for ~25% of the cumulative BC aerosol optical depth in the region for the whole month of January 2020[8]. Katich et al.[5] consolidated measurements of BC emitted from past pyroCb events and concluded that pyroCb-injected smoke is responsible for up to 20% of total organic and BC mass in the lower stratosphere[5]. BC mass loading and its associated climate effects in the stratosphere may become more notable given

[1]Center for Aerosol Science and Engineering, Department of Energy, Environmental and Chemical Engineering, Washington University in St. Louis, St. Louis, MO, USA. [2]Atmospheric, Climate, and Earth Sciences Division, Pacific Northwest National Laboratory, Richland, WA, USA. [3]National Oceanic and Atmospheric Administration (NOAA) Chemical Sciences Laboratory (CSL), Boulder, CO, USA. [4]Department of Atmosphere, Ocean and Earth System Modelling Research, Meteorological Research Institute, Tsukuba, Japan. [5]Department of Chemistry, Colgate University, Hamilton, NY, USA. [6]Cooperative Institute for Research in Environmental Sciences, University of Colorado, Boulder, CO, USA. [7]Present address: BAE Systems, Inc, Boulder, CO, USA. ✉e-mail: chakrabarty@wustl.edu

that pyroCb events are expected to become more common as the magnitude and frequency of large wildfires increase[5,9].

Wildfire-produced BC particles are unique from other aerosols because of their complex shapes and extent of mixing with other aerosol species, which both affect their absorption properties[7,10–13]. Katich et al. [5] examined particle-scale data from pyroCb studies conducted over 13 years and found that large amounts of internally mixed material (henceforth "coatings") are part of a unique fingerprint of pyroCb BC from all cases investigated[5]. The thickness of external coatings on pyrCb BC is distinct from other BC-containing particles in the atmosphere[5,14]. The amount of coating on a BC-containing particle is often quantified using the ratio of coating mass to BC mass ($R_{BC}$), with increased $R_{BC}$ representing increased coating thickness relative to the size of the BC core. There is a large body of research which shows that increased $R_{BC}$ enhances light absorption by BC-containing particles outside of pyroCb clouds[15–21]. However, there are no such studies of light absorption enhancement by BC-containing particles in pyroCb smoke.

Past studies have adopted a top-down approach to predict that absorption by pyroCb-injected smoke in the UTLS causes significant anomalies to stratospheric and global mean temperatures[2,22,23]. Using this approach, aerosol properties are inferred such that modeled pyroCb cloud height and extent match remote sensing measurements[2,22]. This low-resolution methodology suffers from its innate limitations of characterizing the distinct morphologies and microphysical properties of sub micrometer size BC particles within pyroCb clouds. Here, we address this knowledge gap by combining in situ particle-resolved measurements of BC mass and $R_{BC}$ within a pyroCb cloud with bottom-up modeling of BC optical properties to quantify absorption enhancement by pyroCb BC. Lastly, to highlight the unique absorption characteristics of BC in pyroCb smoke, we compare the microphysical and optical properties of BC within the pyroCb cloud to BC from urban sources and from a wildfire which did not produce a pyroCb cloud.

## Results

### Per-particle black carbon mixing state in a pyroCb cloud

A single-particle soot photometer (SP2) provided in situ measurements of pyroCb smoke that was injected by the Williams Flats wildfire in Washington state in August 2019[9]. The SP2 measurements were performed on board the NASA DC-8 research aircraft as part of the Fire Influence on Regional to Global Environments and Air Quality (FIREX-AQ) field campaign[9]. The SP2 measured per-particle BC mass ($m_{BC,i}$) and per-particle $R_{BC}$ ($R_{BC,i}$) during seven intersects of the pyroCb cloud. Figure 1A shows the path of the DC-8 aircraft on August 9 (universal coordinated time, UTC). Backscatter coefficient measured by up-and-down viewing differential absorption lidar−high spectral resolution lidar (DIAL-HSRL) on board the DC-8 aircraft shows that the vertical extent of the pyroCb cloud was ~7.5−10 km, which is distinct from the non-pyroCb smoke layer near the surface (see Supplementary Fig. S1). The thickness, location, and distinct separation of non-pyroCb smoke from the pyroCb cloud is consistent with previous in-situ measurements from a separate pyroCb event[24].

Transmission electron microscopy (TEM) images of particles collected inside the pyroCb cloud show that nearly all BC particles within the pyroCb cloud have external coatings (Fig. 1B). Images with increased resolution further show that the external coatings on BC-containing particles in the pyroCb cloud are extremely thick (Fig. 1C). This is consistent with a previous analysis of particles from pyroCb clouds, which show that over 30% of BC-containing particles have thick external coatings (>200 nm thick), compared with just 3% from non-pyroCb wildfires[5].

To compare the properties of pyroCb BC with BC from other sources, we have utilized SP2 measurements of BC mass-equivalent diameter and $R_{BC,i}$ from a wildfire which did not produce a pyroCb

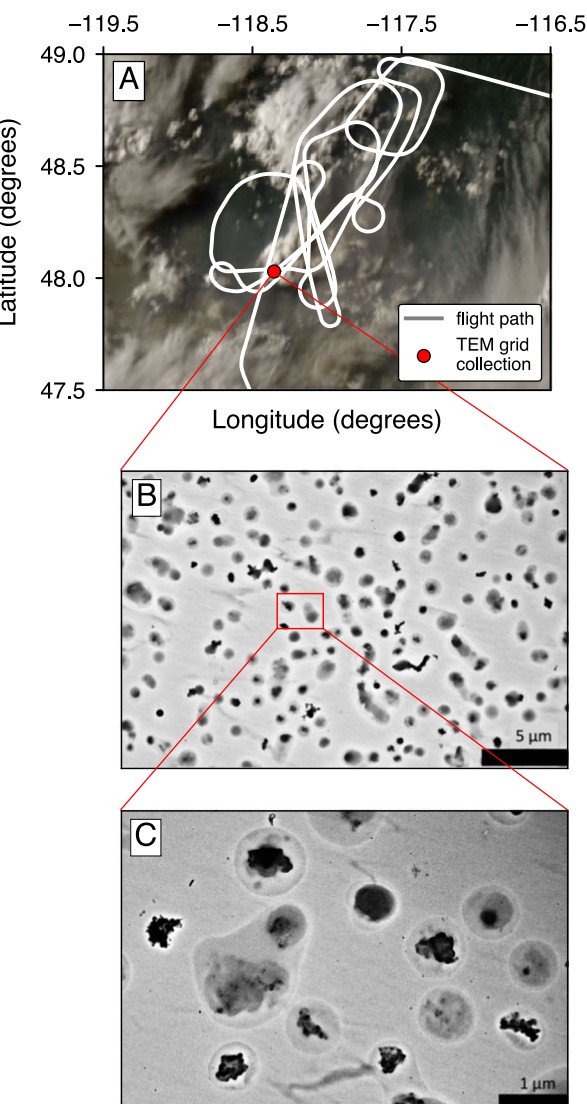

**Fig. 1 | In-situ sampling of black carbon particles in a pyroCb cloud.** **A** Geostationary Operational Environmental Satellites (GOES-17) satellite image of the flight path of the NASA DC-8 aircraft over the Williams Flats wildfire on August 8–9, 2019 (gray). The DC-8 aircraft allowed for in-situ measurements of BC particles during seven intersections of a pyroCb cloud. Particles were collected for TEM imaging during all transects of the flight. A TEM micrograph from one transect is shown in (**B**), and the red point in (**A**) indicates the particle sampling location. TEM images show that a large fraction of BC particles in the pyroCb cloud have external coatings. **C** A zoomed in view of TEM images highlights thick external coatings on BC particles within the pyroCb cloud. TEM images from other pyroCb cloud transects show similarly coated BC and are available from the authors on request.

cloud and from urban sources (Fig. 2, labeled wildfire and urban, respectively). We find that the mass-equivalent BC core diameter of particles within the pyroCb smoke and from wildfire sources are slightly larger than mass-equivalent BC core diameter from urban sources (Fig. 2A). This finding is consistent with a previous study which found that biomass burning produces larger BC particles than urban sources[25]. We also find that the BC core diameter of particles in the pyroCb smoke are not significantly different from wildfire sources, indicating that BC formation mechanisms may not differ significantly between wildfires which drive pyroCb formation and those that do not.

Figure 2B shows the BC mass-weighted distribution of $R_{BC,i}$ for BC particles from urban sources, wildfire sources, and in the pyroCb smoke. We find that there is a large degree of heterogeneity in $R_{BC,i}$ within the pyroCb smoke, which is consistent with previous studies

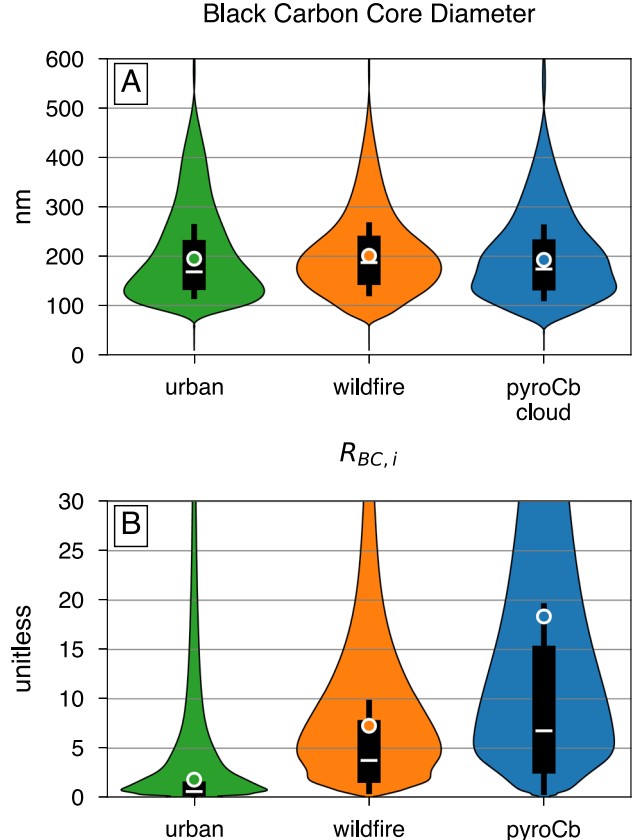

**Fig. 2 | Distribution of black carbon core size and coating amount.** Black carbon (BC) mass-weighted distributions of BC core diameter (**A**) and per-particle $R_{BC}$ ($R_{BC,i}$) (**B**) from urban sources, wildfire, and pyroCb smoke. Overall, BC from wildfires have larger core sizes and thicker coatings than urban BC. However, BC in pyroCb clouds are, on average, more thickly coated than BC from wildfires which do not produce a pyroCb cloud. We have included all BC-containing particles detected by the SP2 in (**A**, **B**). Whiskers represent 5th and 95th percentiles, boxes show inter-quartile ranges, lines represent median values, and circles show average values.

that showed the coating thickness on BC particles in pyroCb clouds can range from <30 nm to 400 nm[5]. We also find that pyroCb BC has significantly larger BC mass-weighted median and average $R_{BC,i}$ than BC from urban and wildfire sources (Fig. 2B). The BC mass-weighted average $R_{BC,i}$ is ~1.8 for urban BC, 7.3 for wildfire BC, and 18.3 for BC within the pyroCb smoke.

Our findings are consistent with Katich et al.'s analysis of pyroCb events over the past 13 years[5]. Their findings unraveled two distinctive features of pyroCb BC: first, they possess extremely thick coatings that differ from non-pyroCb wildfire-sourced BC, and second, the mass concentration ratio of BC to organics in a pyroCb plume—a critical parameter for the modeling plume radiative impacts—remains a constant at ~0.016 ± 0.008. This constant ratio has been observed to sustain over a wide range of altitude, temperature, and plume age, implying that secondary processes of organic aerosol formation is not a dominant coating mechanism for BC in pyroCb plumes post-injection to the UTLS. Informed by these observations, the current consensus holds that coagulation and condensation of low-volatility gases within the strong convective cell of a pyroCb drives the external coating mechanism of BC to a near-stable $R_{BC,i}$. In contrast, BC emissions from urban sources and non-pyroCb wildfires have been observed to undergo large changes in their BC to organics mass ratio both in plume and post-emission because of secondary organic aerosol formation[24,26–28].

Lastly, we find that $R_{BC,i}$ is inversely proportional to BC mass-equivalent core diameter for BC from all sources (Spearman's rank correlation coefficient = -0.49, p « 0.001). Previous studies have also shown that smaller BC particles within populations of BC from urban and wildfire sources contain most of the coating material[29,30]. Our findings provide evidence that this relationship also applies to BC within pyroCb clouds.

### Absorption enhancement by pyroCb black carbon

Several previous studies have shown that absorption increases with increasing $R_{BC}$ due to the so-called "lensing effect", in which light is focused by the coating material into the volume of BC, where it is then absorbed[31–33]. This effect can be quantified using per-particle absorption enhancement factors ($E_{abs,i}$), defined as the ratio of absorption by the coated BC particle to absorption by an uncoated BC core of equal size. However, atmospheric BC particles exist in large populations which exhibit a high degree of heterogeneity in size and mixing state[11]. Therefore, to better understand how the mixing state of populations of pyroCb BC affects its total absorption, we have calculated population-averaged absorption enhancement ($E_{abs}$) and $R_{BC}$ for the pyroCb event measured during FIREX-AQ. For each transect, population-averaged $R_{BC}$ is given by

$$R_{BC} = \frac{\sum_i R_{BC,i} m_{BC,i}}{\sum_i m_{BC,i}}, \tag{1}$$

and population averaged $E_{abs}$ is given by

$$E_{abs} = \frac{\sum_i E_{abs,i} m_{BC,i}}{\sum_i m_{BC,i}}, \tag{2}$$

Here, $R_{BC,i}$ is the ratio of coating to BC mass in particle $i$, $E_{abs,i}$ is modeled absorption enhancement of particle $i$, and $m_{BC,i}$ is the mass of BC in particle $i$. Given that absorption measurements are not available during pyroCb transects, we have used a previously published model which predicts per-particle absorption by coated BC and accounts for the complex shape of BC particles to quantify absorption enhancement within the pyroCb smoke[10]. Supplementary Fig. S2 shows that the model accurately reproduces measured $E_{abs}$ for BC from wildfire and urban sources.

We compare modeled $E_{abs}$ as a function of $R_{BC}$ for BC from urban sources, wildfire sources, and within the pyroCb smoke (Fig. 3A). We find that the mean $E_{abs}$ of pyroCb BC approaches a maximal value of 2.0 at wavelength of 532 nm, meaning that BC in the pyroCb smoke is responsible for up to two times more absorption than were it externally mixed. Figure 3A also shows that $E_{abs}$ of BC in the pyroCb cloud is 1.2–1.7 times higher than $E_{abs}$ of urban BC and up to 1.3 times higher than BC from wildfires which do not produce pyroCbs. Because $E_{abs}$ is given by the BC mass-weighted average of per-particle absorption enhancement ($E_{abs,i}$), the inverse proportionality between $R_{BC,i}$ and BC core size is extremely important for determining $E_{abs}$. Particles with $E_{abs,i} > 2.0$ make up only 3% of the urban BC mass fraction, which causes low $E_{abs}$ for urban BC. On the other hand, particles with $E_{abs,i} > 2.0$ make up 19% of the BC mass fraction in non-pyroCb wildfire BC and 39% of the BC mass fraction in the pyroCb cloud. This results in high $E_{abs}$ for BC emitted from wildfire sources and within the pyroCb cloud. However, it should be noted that the exact behavior of $E_{abs}$ may depend on the source-specific distribution of BC core size and $R_{BC,i}$.

### Upper limits of lensing-induced black carbon absorption enhancement

Our findings also show that population-averaged $E_{abs}$ approaches a maximal value of 2.0 at $R_{BC} \approx 12$, beyond which increased $R_{BC}$ does not result in significant increases to $E_{abs}$ (Fig. 3A), indicating that there is saturation of the lensing effect. This saturation implies that after a certain coating thickness has been achieved, addition of coating on the

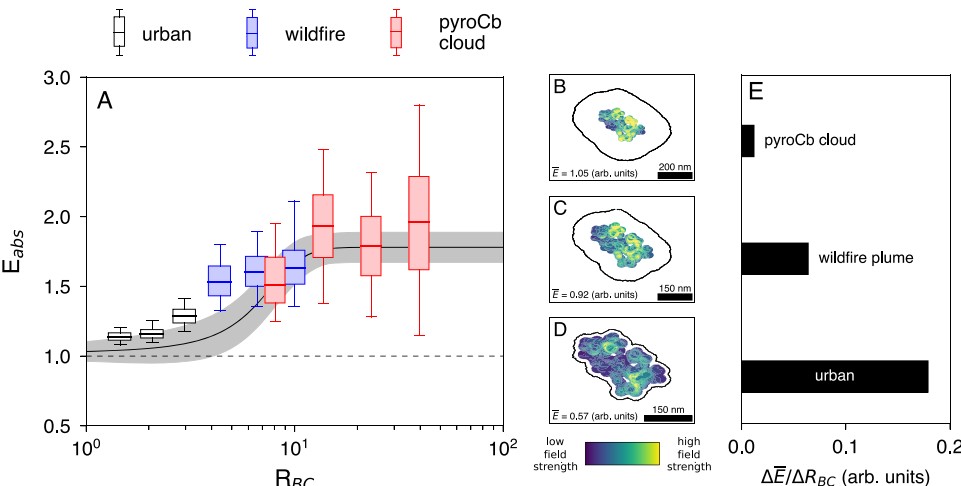

**Fig. 3 | Saturation of lensing-induced absorption enhancement in pyroCb cloud. A** Population-averaged absorption enhancement ($E_{abs}$) at 532 nm wavelength by BC-containing particles from urban sources, wildfire sources, and within a pyroCb plume. We find that extremely thick coatings on BC inside the pyroCb cloud cause higher $E_{abs}$ when compared to BC from urban and wildfire sources. The solid black line in (**A**) shows a sigmoidal fit of $E_{abs}$ measurements for wildfire BC made by ref. 30, with shaded areas representing 95% confidence intervals. Boxes in (**A**) show the inter-quartile range, whiskers represent 5th and 95th percentiles, and lines represent median values. We find that $E_{abs}$ plateaus at ~2.0, providing an upper limit for absorption enhancement by atmospheric BC.

This plateau is visualized in (**B**–**D**), which shows the strength of the induced electric field as light passes through the BC core of a particle which is representative of a particle inside the pyroCb plume (**B**), a non-pyroCb wildfire particle (**C**), and an urban BC particle (**D**). Black contours in (**B**–**D**) represent the silhouette of coatings on the BC particles. We find that increasing $R_{BC}$ beyond ~12 does not lead to significant increases in the average strength of the induced electric field ($\bar{E}$). This is demonstrated in (**E**), which shows the change in $\bar{E}$ per unit change in $R_{BC}$ ($\Delta\bar{E}/\Delta R_{BC}$). We find that $\Delta\bar{E}/\Delta R_{BC}$ decreases significantly as the value of $R_{BC}$ increases, and thus does not lead to significantly increased absorption.

surface of BC does not result in additional light being scattered into the BC core, and subsequently does not result in increased absorption. This phenomenon is demonstrated by Fig. 3B–D, which shows the electric field strength as light passes through a BC core with BC mass-weighted average $R_{BC,i}$ of BC in the pyroCb plume (B), wildfire BC (C), and urban BC (D). The internal electric field strength can be used to quantify the amount of light that reaches the BC core of each particle[34]. Solid black lines in Fig. 3B–D indicate the silhouette of coatings on the BC particles. We find that the average electric field strength over the BC core volume ($\bar{E}$) increases as the amount of coating increases. However, Fig. 3E shows that the change in $\bar{E}$ per change in $R_{BC}$ (obtained by linear regression between panels B and D) decreases as $R_{BC}$ increases. This demonstrates the marginal increase in the amount of light scattered into the BC core as $R_{BC}$ increases, and thusly the marginal increase in $E_{abs,i}$. Because $E_{abs,i}$ approaches a maximal value as $R_{BC,i}$ increases, population-averaged $E_{abs}$ also approaches a maximal value as population-averaged $R_{BC}$ increases. These findings provide a valuable upper limit to atmospheric light absorption by BC from all sources.

Our finding of an asymptotic value of 2.0 is consistent with pre-vious measurements that reported $E_{abs}$ in wildfire plumes to be well described as a sigmoid function of BC coating thickness (shown with black line in Fig. 3A)[30]. Our findings provide further evidence that the sigmoid function reported by Lee et al.[30] is suitable for estimating $E_{abs}$ for BC from urban and wildfire sources and may also be suitable for estimating $E_{abs}$ in pyroCb clouds. The asymptotic value of $E_{abs} \approx 2.0$ obtained from modeling in this work and observation by Lee et al. are in contrast with traditional core-shell Mie theory calculations which predict $E_{abs}$ of up to 2.5. Overestimation of $E_{abs}$ by traditional models has been a persistent issue in large-scale radiative transfer models, and our findings provide further evidence that more complex representa-tion of BC mixing state is needed to accurately predict $E_{abs}$[35].

## Discussion

Overall, this work shows that BC-containing particles in pyroCb clouds are distinct from urban and wildfire BC and provides evidence of an

upper limit to lensing-induced BC absorption enhancement. This work also builds on previous studies of the relationship between $R_{BC}$ and $E_{abs}$[10,15,17–19,29,32,36–38]. While these studies conclusively show that $E_{abs}$ increases as $R_{BC}$ increases, they were focused on BC from urban and non-pyroCb biomass burning sources. Our work focuses on this rela-tionship for pyroCb BC and shows that $E_{abs}$ approaches a maximum value of ~2.0. As per-particle $R_{BC}$ reaches ~12, further addition of coating material does not result in significant increases in per-particle $E_{abs}$ and as a result does not cause significant increases to population-averaged $E_{abs}$ ($\Delta E_{abs}/\Delta R_{BC} \to 0$ as $R_{BC} \to \infty$). This finding is supported by modeling results which show that the increase in internal electric field strength per unit increase in per-particle $R_{BC}$ decreases as per-particle $R_{BC}$ increases, and by direct measurements which also observed an asymptotic value of $E_{abs}$ at high $R_{BC}$[30].

A limitation of this study is its focus on a singular fresh pyroCb event and its comparison with a limited dataset of urban and wildfire events. The freshness of the pyroCb plume investigated here could raise a question regarding the BC particles in the plume being at or near their maximal coating thickness, and that further aging of the plume could result in a loss of coating mass and decreased $E_{abs}$[13]. However, the comprehensive study by ref. 5 showed that that the mixing states of pyroCb BC remains relatively unchanged over a period of months in the UTLS[5]. Photochemical aging and coating processes that are often the result of secondary aerosol formation by the organic gases associated with fire emissions are not dominant in the stratosphere[5]. Hence, the major finding from this study, that is, $E_{abs}$ approaching 2, is expected to hold valid for both in-cloud and injected pyroCb BC in the stratosphere.

Maximal $E_{abs}$ of ~2.0 has implications for the climate impacts of BC-containing particles. It is common for large-scale radiative transfer models to use a simplifying assumption that all BC-containing particles in a population have identical $R_{BC}$. Recent studies have shown that optics models that use this assumption overestimate $E_{abs}$ for urban and wildfire BC[11,30]. Our work adds to a growing body of evidence that $E_{abs}$ plateaus at high $R_{BC}$[10,30,36], as well as underscores the fact that simpli-fying model assumptions may overestimate $E_{abs}$ for BC in pyroCb

clouds. Overestimation of $E_{abs}$ by pyroCb BC will in turn lead Earth system and chemical transport models to overestimate warming caused by pyroCb BC. A recent study compared measured $E_{abs}$ to predictions from nine state-of-the-art Earth system and chemical transport models and showed that warming caused by biomass burning aerosols is systematically overestimated[35]. Our findings indicate that these models may also overestimate warming by pyroCb BC in a similar manner. The implications of our findings are particularly notable given that pyroCb events are responsible for almost a quarter fraction of the BC and organic mass in the UTLS[5,8]. The prevalence of pyroCb BC in the UTLS, coupled with our findings of enhanced pyroCb BC absorption show that pyroCb BC could have a large influence on UTLS temperatures, which have a strong impact on global radiative forcing, atmospheric circulation, and stratospheric chemistry[37,38]. Our findings highlight the need for more detailed observation and modeling efforts of pyroCb clouds, especially given that this century has already seen two record-setting pyroCb outbreaks, and the frequency of large-scale biomass burning events is expected to increase in the future[2,39].

## Methods

### FIREX-AQ sampling of pyroCb and wildfire plumes

This study utilized measurements of pyroCb smoke stemming from the Williams Flats wildfire in eastern Washington, USA. This fire produced three separate pyroCb clouds which were intercepted after evaporation by the NASA DC-8 aircraft on August 9, 2019 (UTC)[9]. PyroCb plume intercepts were identified by elevated CO concentrations, a marker of biomass burning emissions (see Fig. S3)[9]. The DC-8 intersected the pyroCb cloud 11 total times, but >100 BC particles were detected by the SP2 in 7 transects. We have included data from only these transects in our analysis.

This study also used measurements from inside a wildfire plume stemming from the Castle and Ikes wildfires in northern Arizona, USA. The DC-8 aircraft intercepted those plumes on August 12 and August 13, 2019 (UTC) (see Fig. S4). While the DC-8 intercepted plumes from several wildfires during FIREX-AQ, we have included data from only the Castle and Ikes wildfires. This is due to detailed knowledge of the contribution of non-BC particles to total light absorption within these fires[40]. However, the microphysical properties of BC within the Castle and Ikes wildfire plumes do not differ significantly from the microphysical properties of BC measured in all wildfire plumes. Therefore, we do not expect that exclusion of SP2 measurements from other wildfire plumes alters the findings of this work (see Table S1).

The $R_{BC}$ of particles within each plume was measured by a SP2 on board the DC-8 aircraft[41,42]. The SP2 detected the mass of BC in each particle and the total optical size of each particle, which is then converted to per-particle non-BC coating thickness[42]. We have converted per-particle BC mass and coating thickness to per-particle $R_{BC}$ assuming a core-shell arrangement and densities of 1.8 and 1.2 g/cm³ for BC and coating, respectively[7]. The distributions of measured $R_{BC}$ and BC core size are shown in Fig. 2.

Aerosol particle samples were collected on TEM grids within the pyroCb cloud using an impactor sampler (AS-24W, Arios Inc) on board the DC-8 aircraft. The samples were analyzed by TEM (JEM-1400, JEOL) to measure their mixing states as shown in Fig. 1.

### Mixing state and absorption by urban BC

We utilized previously published datasets of SP2-measured per-particle BC mass and non-BC coating thickness from the 2013 Studies of Emissions and Atmospheric Composition, Clouds and Climate Coupling by Regional Surveys (SEAC4RS) campaign. We have utilized SP2 measurements on board a DC-8 aircraft on September 23, 2013. We have included data from 16:45–17:00 UTC and 23:00–23:35 UTC, which cover sampling near the Houston, TX shipping channel and over California, respectively[43]. We have converted per-particle BC mass and

coating thickness to per-particle $R_{BC}$ assuming a core-shell arrangement and densities of 1.8 and 1.2 g/cm³ for BC and coating. The distributions of measured $R_{BC}$ and BC core size are shown in Fig. 2.

Measured total light absorption (Mm⁻¹) was obtained from PSAP measurements at wavelength = 532 nm. We assume that BC is responsible for 94% of absorption at 532 nm[44], and correct filter-based absorption measurements to account for measurement artifacts[45]. We have taken 10-min averages of total light absorption and BC mass concentration (g/m³, measured by SP2). The BC mass absorption coefficient is then given by the ratio of light absorption per BC mass concentration (m²/g). Measured $E_{abs}$ is then calculated by dividing the measured BC mass absorption coefficient by the mass absorption coefficient of pure BC at 532 nm (7.75 m²/g)[6]. A comparison of measured and modeled absorption enhancement for urban BC can be found in Fig. S2. We find that the modeled absorption enhancement accurately replicates the measured absorption enhancement, and therefore only shows modeled values in Fig. 3.

### Measured absorption by wildfire BC

Total light absorption (Mm⁻¹) at wavelength = 664 nm was measured by a photoacoustic spectrometer on board the DC-8 aircraft. Detailed analysis of particles from the Castle and Ikes wildfires has estimated that organics are responsible for 45% of total light absorption at 664 nm[40]. Therefore, we conclude that BC is responsible for the remaining 55% of total absorption. We have taken 10-min averages of total light absorption and BC mass concentration (g/m³, measured by SP2). The BC mass absorption coefficient is then given by the ratio of light absorption per BC mass concentration (m²/g). Measured $E_{abs}$ is then calculated by dividing the measured BC mass absorption coefficient by the mass absorption coefficient of pure BC at 664 nm (6.21 m²/g)[6]. A comparison of measured and modeled absorption enhancement for wildfire BC can be found in Fig. S2. We find that the modeled absorption enhancement accurately replicates the measured absorption enhancement, and therefore only shows modeled values in Fig. 3.

### Modeled absorption by BC in a pyroCb cloud

While measurements of total absorption were available for BC from urban and wildfire sources, there are no measurements of absorption within the pyroCb cloud. Therefore, we have used previously published scaling laws for BC light absorption to quantify $E_{abs}$ in the pyroCb plume[10]. This is achieved by firstly using the scaling relationships to calculate the absorption cross-section of each BC-containing particle measured by the SP2 in the pyroCb cloud ($C_{abs,coated}$). The scaling relationships are then used to calculate the absorption cross-section of a pure BC particle with equivalent BC mass ($C_{abs,bare}$). We then take the ratio of $C_{abs,coated}$ to $C_{abs,bare}$ to give per-particle absorption enhancement ($E_{abs,i}$), which is then used to calculate $E_{abs}$ using Eq. 2. The optics model used in this work assumes that BC monomers have diameters of 40 nm, refractive index of 1.95 + 0.79i, and are coated with non-absorbing organics[6,7]. The absorption cross-section predicted by the model approximates the orientation-averaged cross-section obtained from the Amsterdam Discrete Dipole Approximation algorithm (ADDA)[46]. Previous work has shown that this model replicates measurements of BC $E_{abs}$ from numerous studies[10] and replicates measured $E_{abs}$ for the urban and wildfire BC investigated in this study (see Fig. S2). Figure 3 shows modeled 10-min average values of $E_{abs}$ at wavelength of 532 nm for the pyroCb cloud intersects during FIREX-AQ.

### Calculation of internal electric fields

Figure 3 shows modeled induced internal electric fields for three particles that are representative of urban, wildfire, and pyroCb BC. All particles have BC mass of ≈7.0 fg (mass-weighted average of BC from all sources). Figure 3d represents an urban BC particle with $R_{BC}$ ≈ 1.8 (BC mass-weighted average value). Figure 3c represents a wildfire BC

particle with $R_{BC} \approx 7.3$. Figure 3b represents a BC particle from inside a pyroCb cloud with $R_{BC} \approx 18.3$. It is well established that freshly emitted BC exists as fractal aggregates with fractal dimension $(D_f) = 1.8$[14,47]. Previous studies have shown that addition of external coatings can cause significant restructuring of BC to more collapsed shapes $(D_f > 1.8)$[48]. To account for core restructuring, we have assumed that all particles have have $D_f \approx 2.5$. We used a dense diffusion limited cluster-cluster aggregation model to simulate the complex shape of each BC particle shown in Fig. 3B–D[47,49]. The diameter of BC monomers was assumed to be 40 nm[6]. Once the BC core shape was generated, coating was then simulated on the surface of BC until the desired $R_{BC}$ was achieved.

The internal electric field of each particle was then calculated using ADDA[46]. The refractive index of BC is assumed to be $1.95 + 0.79i$ and the refractive index of the coating is assumed to be $1.55 + 0.00i$[6,7]. The refractive index of the coating material is representative of organics, which has been shown to be the predominant coating material in biomass burning and pyroCb BC[2,5]. Calculations are performed at wavelength of 532 nm. The ADDA algorithm operates by discretizing particles into sub-volumes which are significantly smaller than the wavelength of incident light. To minimize errors in the internal field calculations, it is recommended that the wavelength of incident light be at least 10 times the size of each sub-volume[46]. Our calculations were performed with the wavelength of incident light greater than 150 times the size of each sub-volume, so minimal errors are expected in calculating the internal electric field strength.

## Data availability

All raw data used for this study are available for public use on NASA's FIREX-AQ data repository: https://www-air.larc.nasa.gov/cgi-bin/ArcView/firexaq. The transmission electron microscopy images are available from the corresponding authors upon request.

## Code availability

Python code used to calculate BC optical properties is described in detail in ref. 10 and is available for download from https://pypi.org/project/pyBCabs/.

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

## Acknowledgements

This research has been supported by the National Aeronautics and Space Administration (grant nos. 80NSSC18K1414 and NNH20ZDA001N-ACCDAM), the National Oceanic and Atmospheric Administration (grant no. NA16OAR4310104), the National Science Foundation (grant nos. AGS-1455215 and AGS-1926817), the US Department of Energy (grant no. DE-SC0021011), and the Simons Foundation's Mathematics and Physical Sciences division. L.F. was supported by the U.S. Department of Energy (DOE) Atmospheric System Research (ASR) program via the Integrated Cloud, Land-Surface, and Aerosol System Study (ICLASS) Science Focus Area. Additional support was provided by the Laboratory Directed Research and Development program (Linus Pauling Distinguished Postdoctoral Fellowship Program). Pacific Northwest National Laboratory is operated for DOE by Battelle Memorial Institute under contract DE-AC05-76RL01830.

## Author contributions

P.B.: Conceptualization, Methodology, Software, Data curation, Writing—original draft. J.K.: Conceptualization, Methodology, Data curation, Writing—original draft. J.P.S.: Data curation, Methodology, Writing—review and editing. K.A.: Data curation, Writing—review and editing. L.F.: Methodology, Writing—review and editing. A.E.P.: Data curation, Writing—review and editing. J.M.K.: Data curation, Writing—review, and editing. R.K.C.: Conceptualization, Methodology, Writing—original draft.

## Competing interests

The authors declare no competing interests.
