## [Peer Review File · Nature Communications]

Light Absorption Enhancement by Black Carbon in a Pyrocumulonimbus CloudREVIEWER COMMENTS

Reviewer #1 (Remarks to the Author):

The manuscript by Beeler et al. offers an analysis focused on a case study of pyrocumulonimbus (pyroCB) clouds, highlighting the observation of black carbon (BC) light absorption enhancement, which asymptotically approaches a factor of 2.0. The study's exploration of the extended range of E_{abs} (the enhancement factor) reaching this asymptote presents an intriguing aspect. However, the significance of BC-induced E_{abs} enhancement in the context discussed seems to be overstated. Given that the BC mass fraction in pyroCB plumes rapidly attains a level of 1-2%, an enhancement factor of 2.0 in light absorption does not markedly amplify the overall impact, especially when considering the inherent variability in BC mass fractions within such plumes.

Furthermore, the analysis conducted has a notable resemblance to the findings and asymptotic value of 2.0 reported by Lee (2022). Including a discussion on this similarity, or incorporating their data into Figure 3, would provide valuable insight and contribute to a more comprehensive understanding of the subject matter.

Lee, J. E., Gorkowski, K., Meyer, A. G., Benedict, K. B., Aiken, A. C., & Dubey, M. K. (2022). Wildfire Smoke Demonstrates Significant and Predictable Black Carbon Light Absorption Enhancements. *Geophysical Research Letters*, 49(14). <https://doi.org/10.1029/2022GL099334>

Reviewer #2 (Remarks to the Author):

Review comments to the “Light Absorption Enhancement by Black Carbon in a Pyrocumulonimbus Cloud” by Beeler et al.

This study elucidated for the first time the microphysical properties (size distribution, coating thickness) of BC-containing particles in pyroCb and estimated their absorption enhancement factor. The results are useful for parameterization of radiative properties of BC injected by pyroCb events, which likely accounts for ~20% of background BC concentration in the UTLS. I can recommend the publication of this paper in *Nature Communications* if the authors appropriately take into account the following suggestions.

Major comments:

1. Although the results are important for refining the performance of climate modeling, I'm wondering if the current manuscript would provide essentially new insights or breakthroughs that attract interest from a broad range of scientific communities (outside the community of atmospheric aerosol). I need to say that the expected range of audience attraction is narrower as compared with the previous closely related research (e.g.,

<https://www.science.org/doi/10.1126/science.add3101>). I would recommend adding some discussion on the implications of global radiative effects, rather than focusing only on the details of the single-particle optics.

2. The current manuscript lacks some explanations necessary to support the accuracy of the light-scattering calculations. I didn't find the specification of wavelengths, number of monomers, particle's orientations, and validity of the assumption of the refractive index.

Minor/Individual comments:

Figure 1. I cannot see the Fringe Track (gray line) in the figure.

Section 2, 2nd paragraph: "Transmission electron microscopy (TEM) images of particles collected inside the pyroCb cloud show that nearly all BC-containing particles within the pyroCb cloud have external coatings (Figure 1b)." -> It would be a bit weird to say "BC-containing particles have external coatings" because each BC-containing particle itself includes coating materials.

Figure 2B: Which range of BC core diameter was considered for calculating these R_{BC} statistics of BC-containing particles? Add descriptions in the figure caption or main text.

Section 2, 4th paragraph: "Figure 2b shows the BC mass-weighted distribution of $R_{BC,i}$ for BC particles from urban sources, wildfire sources, and within the pyroCb cloud." -> Define the "BC mass-weighted distribution of $R_{BC,i}$ ". Does the result depend on the detectable BC size range of the SP2?

Section 2, 4th paragraph: "is inversely related to" -> I would suggest using "is inversely proportional to" for clarity if the relationship is linear.

Figure 3: Which wavelength did you assume for these calculations?

L212: "average electric field strength (\bar{E})" -> Is this the average over the BC core volume or the average over the entire volume of BC-containing particle? Please clarify.

L322-343: Please specify the assumed wavelengths for these DDA calculations.

L336: "The refractive index of BC is assumed to be $1.95 + 0.79i$ " -> I wonder if there are strong reasons supporting the validity of this assumption. It was reported that Bond and Bergstrom's recommendation of $1.95 + 0.79i$ underestimates the MAC of bare BC by about 30% (compared with direct MAC measurements) at ~532 nm wavelength. Please see more recent papers on BC refractive index:

<https://doi.org/10.1080/02786826.2019.1676878>, <https://doi.org/10.1080/02786826.2023.2202243>

L337: "the refractive index of the coating is assumed to be $1.55 + 0.00i$." -> Does this value represent the refractive index of organic matter from wildfire? Please provide some pieces of

evidence.

Reviewer #3 (Remarks to the Author):

“Light Absorption Enhancement by Black Carbon in a Pyrocumulonimbus Cloud” by Beeler et al., examines the properties and radiative effects of black carbon containing aerosols from pyroCb events and how they are similar to black carbon from other sources. They find that pyroCb aerosols tend to have thicker coatings and higher absorption enhancement factors than other analogous aerosols. Understanding the properties of pyroCb events is of great interest with many large, even volcanic sized, pyroCb events happening in the past few years affecting global air quality, stratospheric composition, and the planet’s radiative balance. Showing that pyroCb aerosols are unique from other biomass burning aerosols in their coating diameters and radiative properties would help in better modeling their effects and accounting for their impact on the radiative balance of the planet. The submitted manuscript is clearly written and presents an interesting case study but lacks the statistical basis given its small sample size to support some of the broad conclusions reached. The manuscript may be suitable for publication after presenting the results in better context.

Main Points:

1. The main findings from the paper are derived from one pyroCb case. The Williams Flat case is interesting and noteworthy, but the authors would be wise to present their findings with this limitation in mind. Likewise the non-pyroCb dataset is derived from two fires in Arizona. Given the dearth of in-situ airborne observations of pyroCb aerosols, the results presented here are of high interest, but the broad comparisons and conclusions should be tempered with this limitation in mind (e.g., line 241). Also of note, these plumes were relatively fresh. Some studies also suggest that the optical properties of biomass burning/pyroCb aerosols change with age suggesting that these plumes and their aerosols may change coatings-wise with time (Hodshire et al., 2019; Christian et al., 2020).
2. In my opinion, the paper is often missing the “why.” For example, when describing the previous studies of BC aerosol coatings, a sentence or two to describe how these coatings arise and what they are made of would be appropriate. When describing the observed differences between pyroCb aerosols and those from other wildfires and other BC sources, providing a physical mechanism that would lead to these differences would create a more satisfying discussion. Without describing, hypothesizing, or reiterating the physical mechanisms that lead to these differences, readers may be tempted to chalk up these differences to the small sample size.

Minor/Editorial Points:

1. L40: Consider rewording the sentence to make clear that the convection is related to the heat generated by the wildfire, not heat in general.
2. L52: What types of previous studies showed this? Modeling, in situ, etc.?

References:

Hodshire, A. L., Akherati, A., Alvarado, M. J., Brown-Steiner, B., Jathar, S. H., Jimenez, J. L., Kreidenweis, S. M., Lonsdale, C. R., Onasch, T. B., Ortega, A. M., and Pierce, J. R. "Aging Effects on Biomass Burning Aerosol Mass and Composition: A Critical Review of Field and Laboratory Studies", *Environmental Science & Technology*. 2019 53 (17), 10007-10022. DOI: 10.1021/acs.est.9b02588

Christian, K., Yorks, J., and Das. S. "Differences in the Evolution of Pyrocumulonimbus and Volcanic Stratospheric Plumes as Observed by CATS and CALIOP Space-Based Lidars" *Atmosphere* 2020 11, 1035; doi:10.3390/atmos11101035

Response to Reviewers' Comments

The reviewers' comments are in regular black font. Our response is in regular blue font. Changes made to the manuscript are in *italics blue* font color.

Reviewer #1 (Remarks to the Author):

The manuscript by Beeler et al. offers an analysis focused on a case study of pyrocumulonimbus (pyroCB) clouds, highlighting the observation of black carbon (BC) light absorption enhancement, which asymptotically approaches a factor of 2.0. The study's exploration of the extended range of E_{abs} (the enhancement factor) reaching this asymptote presents an intriguing aspect. However, the significance of BC-induced E_{abs} enhancement in the context discussed seems to be overstated. Given that the BC mass fraction in pyroCB plumes rapidly attains a level of 1-2%, an enhancement factor of 2.0 in light absorption does not markedly amplify the overall impact, especially when considering the inherent variability in BC mass fractions within such plumes.

We thank the reviewer for making this astute observation. Indeed, it is important to provide the readers with a brief overview of the overall impact associated with a ~2% (or 0.3 Tg) mass injection of pyroCb BC into the stratosphere. In the revised manuscript, we have added a new line in the abstract:

Injected BC warms the local stratospheric air and consequently perturbs the spatial distributions of ozone and water vapor.

We also added a new paragraph to the Introduction section (see lines 46 – 59)

“BC makes up around 2% of the total particle mass in pyroCb clouds, yet the contribution of BC to UTLS light absorption and associated warming is significant⁵. Yu et al. (2019) reported BC from an August 2017 pyroCb event over western Canada to have warmed the local UTLS air by up to 7 K². Because of this warming, large spatial anomalies in ozone and water vapor concentrations were observed corresponding to the plume location in the UTLS. Owing to its long atmospheric lifetime, BC from pyroCb events not only enhances shortwave radiation absorption, but also impacts the spatial distributions of UTLS atmospheric trace species. Tang et al. (2021) reported that a single episode of pyroCb wildfire in Australia alone accounted for ~25% of the cumulative BC aerosol optical depth in the region for the whole month of January 2020⁸. Katich et al. (2023) consolidated measurements of BC emitted from past pyroCb events and concluded that pyroCb-injected smoke is responsible for up to ~20% of total organic and BC mass in the UTLS⁵. BC mass loading and its associated climate effects in the stratosphere may become more notable given that pyroCb events are expected to become more common as the magnitude and frequency of large wildfires increase^{5,9}.”

References:

Yu, P. et al. Black carbon lofts wildfire smoke high into the stratosphere to form a persistent plume. *Science* **365**, 587–590 (2019).

Katich, J. et al. Pyrocumulonimbus affect average stratospheric aerosol composition. *Science* **379**, 815–820 (2023).

Tang, W., Lloret, J., Weis, J. *et al.* Widespread phytoplankton blooms triggered by 2019–2020 Australian wildfires. *Nature* **597**, 370–375 (2021).

Peterson, D. A. *et al.* Measurements from inside a thunderstorm driven by wildfire: The 2019 FIREX-AQ field experiment. *Bull. Am. Meteorol. Soc.* **103**, E2140–E2167 (2022).

Furthermore, the analysis conducted has a notable resemblance to the findings and asymptotic value of 2.0 reported by Lee (2022). Including a discussion on this similarity, or incorporating their data into Figure 3, would provide valuable insight and contribute to a more comprehensive understanding of the subject matter.

We appreciate this suggestion and have added the sigmoidal fit provided by Lee (2022) in Figure 3. We have also added a paragraph (lines 240-249) discussing our results in the context of this study.

Lee, J. E., Gorkowski, K., Meyer, A. G., Benedict, K. B., Aiken, A. C., & Dubey, M. K. (2022). Wildfire Smoke Demonstrates Significant and Predictable Black Carbon Light Absorption Enhancements. *Geophysical Research Letters*, 49(14). <https://doi.org/10.1029/2022GL099334>

This reference is added to the revised manuscript.

Reviewer #2 (Remarks to the Author):

Review comments to the “Light Absorption Enhancement by Black Carbon in a Pyrocumulonimbus Cloud” by Beeler et al.

This study elucidated for the first time the microphysical properties (size distribution, coating thickness) of BC-containing particles in pyroCb and estimated their absorption enhancement factor. The results are useful for parameterization of radiative properties of BC injected by pyroCb events, which likely accounts for ~20% of background BC concentration in the UTLS. I can recommend the publication of this paper in *Nature Communications* if the authors appropriately take into account the following suggestions.

Major comments:

1. Although the results are important for refining the performance of climate modeling, I’m wondering if the current manuscript would provide essentially new insights or breakthroughs that attract interest from a broad range of scientific communities (outside the community of atmospheric aerosol). I need to say that the expected range of audience attraction is narrower as compared with the previous closely related research (e.g., <https://www.science.org/doi/10.1126/science.add3101>). I would recommend adding some discussion on the implications of global radiative effects, rather than focusing only on the details of the single-particle optics.

We appreciate this feedback and have expanded on the radiative implications of our findings in the discussion (see lines 276 – 294):

“Maximal E_{abs} of ~ 2.0 has implications for the climate impacts of BC-containing particles. It is common for large-scale radiative transfer models to use a simplifying assumption that all BC-containing particles in a population have identical R_{BC} . Recent studies have shown that optics models that use this assumption overestimate E_{abs} for urban and wildfire BC^{11,29}. Our work adds to a growing body of evidence that E_{abs} plateaus at high R_{BC} ^{10,29,35}, as well as underscores the fact that simplifying model assumptions may overestimate E_{abs} for BC in pyroCb clouds. Overestimation of E_{abs} by pyroCb BC will in turn lead Earth system and chemical transport models to overestimate warming caused by pyroCb BC. A recent study compared measured E_{abs} to predictions from nine state-of-the-art Earth system and chemical transport models and showed that warming caused by biomass burning aerosols is systematically overestimated³⁴. Our findings indicate that these models may also overestimate warming by pyroCb BC in a similar manner. The implications of our findings are particularly notable given that pyroCb events are responsible for almost a quarter fraction of the BC and organic mass in the UTLS^{5,8}. The prevalence of pyroCb BC in the UTLS, coupled with our findings of enhanced pyroCb BC absorption show that pyroCb BC could have a large influence on UTLS temperatures, which have a strong impact on global radiative forcing, atmospheric circulation, and stratospheric chemistry^{36,37}. Our findings highlight the need for more detailed observation and modeling efforts of pyroCb clouds, especially given that this century has already seen two record-setting pyroCb outbreaks, and the frequency of large-scale biomass burning events is expected to increase in the future^{2,38}.”

2. The current manuscript lacks some explanations necessary to support the accuracy of the light-scattering calculations. I didn't find the specification of wavelengths, number of monomers, particle's orientations, and validity of the assumption of the refractive index.

The optics model used in this work is based on orientation-averaged simulations using the Amsterdam Discrete Dipole Approximation algorithm. A previous publication provides detailed comparison of model predictions with measurements and shows that the model accurately predicts the absorption cross-section of coated aggregates (see Beeler and Chakrabarty 2022). The wavelength presented in this work is 532 nm. We have provided these details on lines 209 and 361 – 369.

“We find that E_{abs} of BC in the pyroCb cloud approaches a maximal value of 2.0 at wavelength of 532 nm, meaning that BC in the pyroCb cloud is up to two times more absorbing than uncoated BC.”

“The optics model used in this work assumes that BC monomers have diameters of 40 nm, refractive index of $1.95 + 0.79i$, and are coated with non-absorbing organics^{6,7}. The absorption cross-section predicted by the model approximates the orientation-averaged cross-section obtained from the Amsterdam Discrete Dipole Approximation algorithm⁴⁵. Previous work has shown that this model replicates measurements of BC E_{abs} from numerous studies¹⁰ and replicates measured E_{abs} for the urban and wildfire BC investigated in this study (see Figure S4). Figure 3 shows modeled ten-minute average values of E_{abs} at wavelength of 532 nm for the pyroCb cloud intersects during FIREX-AQ.”

Minor/Individual comments:

Figure 1. I cannot see the Fright Track (gray line) in the figure.

This may be an issue with the graphics format. We have switched formats and hope this is corrected in the updated manuscript.

Section 2, 2nd paragraph: “Transmission electron microscopy (TEM) images of particles collected inside the pyroCb cloud show that nearly all BC-containing particles within the pyroCb cloud have external coatings (Figure 1b).” -> It would be a bit weird to say “BC-containing particles have external coatings” because each BC-containing particle itself includes coating materials.

We have reworded this sentence for clarity.

Figure 2B: Which range of BC core diameter was considered for calculating these R_{BC} statistics of BC-containing particles? Add descriptions in the figure caption or main text.

We have included the entire range of BC core diameters detected by the SP2 in our analysis. The smallest BC core detected was approximately 65 nm in diameter and the largest was 485 nm.

While the SP2 can introduce errors in coating thickness measurements for small particles, these do not contribute significantly to the properties of the population, which are weighted by BC mass. We have included this information in the caption of Figure 2.

Section 2, 4th paragraph: “Figure 2b shows the BC mass-weighted distribution of $R_{BC,i}$ for BC particles from urban sources, wildfire sources, and within the pyroCb cloud.” -> Define the “BC mass-weighted distribution of $R_{BC,i}$ ”. Does the result depend on the detectable BC size range of the SP2?

Figure 2b shows mass distribution function of $R_{BC,i}$. For example, BC particles with $R_{BC,i}$ of 5 make up a majority of the total BC mass in wildfire plumes. The result is weakly dependent on the detectable BC size range of the SP2. Because small BC cores tend to be more thickly coated, their exclusion leads to slight decreases in population-average R_{BC} and E_{abs} . However, these particles do not make up a significant fraction of the BC mass and therefore the effect on BC-mass weighted R_{BC} and E_{abs} is minimal.

Section 2, 4th paragraph: “is inversely related to” -> I would suggest using “is inversely proportional to” for clarity if the relationship is linear.

This has been corrected in the revised text.

Figure 3: Which wavelength did you assume for these calculations?

All calculations are performed at wavelength of 532 nm. This has been clarified in the revised text, see lines 209 and 361 – 369.

L212: “average electric field strength (\bar{E})” -> Is this the average over the BC core volume or the average over the entire volume of BC-containing particle? Please clarify.

We present the average electric field strength over the BC core volume. This has been clarified in the revised text on line 231 – 232.

L322-343: Please specify the assumed wavelengths for these DDA calculations.
We have included the wavelength for DDA calculations on line 209 and 361 – 369 .

L336: “The refractive index of BC is assumed to be $1.95 + 0.79i$ ” -> I wonder if there are strong reasons supporting the validity of this assumption. It was reported that Bond and Bergstrom’s recommendation of $1.95 + 0.79i$ underestimates the MAC of bare BC by about 30% (compared with direct MAC measurements) at ~532 nm wavelength. Please see more recent papers on BC refractive index:

<https://doi.org/10.1080/02786826.2019.1676878>, <https://doi.org/10.1080/02786826.2023.2202243>

The optics model used in this work was developed using DDA simulations with BC refractive index of $1.95+0.79i$. This model has been validated using several lab and ambient measurements of MAC. We also compare E_{abs} calculated using the optics model with measured E_{abs} from urban and wildfire sources and find that the model accurately calculates E_{abs} . We have elected to use the same BC refractive index for internal electric field calculations for consistency.

L337: “the refractive index of the coating is assumed to be $1.55 + 0.00i$.” -> Does this value represent the refractive index of organic matter from wildfire? Please provide some pieces of evidence.

This value represents organic coatings. We have added this detail with justification on lines 385 – 388.

“The refractive index of the coating material is representative of organics, which has been shown to be the predominant coating material in biomass burning and pyroCb BC^{2,5}. Calculations are performed at wavelength of 532 nm.”

References: Yu, P. et al. Black carbon lofts wildfire smoke high into the stratosphere to form a persistent plume. *Science* **365**, 587–590 (2019).

Katich, J. et al. Pyrocumulonimbus affect average stratospheric aerosol composition. *Science* **379**, 815–820 (2023).

Reviewer #3 (Remarks to the Author):

“Light Absorption Enhancement by Black Carbon in a Pyrocumulonimbus Cloud” by Beeler et al., examines the properties and radiative effects of black carbon containing aerosols from pyroCb events and how they are similar to black carbon from other sources. They find that pyroCb aerosols tend to have thicker coatings and higher absorption enhancement factors than other analogous aerosols. Understanding the properties of pyroCb events is of great interest with many large, even volcanic sized, pyroCb events happening in the past few years affecting global air quality, stratospheric composition, and the planet’s radiative balance. Showing that pyroCb aerosols are unique from other biomass burning aerosols in their coating diameters and radiative properties would help in better modeling their effects and accounting for their impact on the

radiative balance of the planet. The submitted manuscript is clearly written and presents an interesting case study but lacks the statistical basis given its small sample size to support some of the broad conclusions reached. The manuscript may be suitable for publication after presenting the results in better context.

Main Points:

1. The main findings from the paper are derived from one pyroCb case. The Williams Flat case is interesting and noteworthy, but the authors would be wise to present their findings with this limitation in mind. Likewise the non-pyroCb dataset is derived from two fires in Arizona. Given the dearth of in-situ airborne observations of pyroCb aerosols, the results presented here are of high interest, but the broad comparisons and conclusions should be tempered with this limitation in mind (e.g., line 241). Also of note, these plumes were relatively fresh. Some studies also suggest that the optical properties of biomass burning/pyroCb aerosols change with age suggesting that these plumes and their aerosols may change coatings-wise with time (Hodshire et al., 2019; Christian et al., 2020).

We thank the reviewer for raising this point. In the discussion section of the revised manuscript, we have added the following paragraph outlining the limitation, as well as the universality of our findings (lines 265 –274):

“A limitation of this study is its focus on a singular fresh pyroCb event and its comparison with a limited dataset of urban and wildfire events. The freshness of the pyroCb plume investigated here could raise a question regarding the BC particles in the plume being at or near their maximal coating thickness, and that further aging of the plume could result in a loss of coating mass and decreased E_{abs}^{13} . However, the comprehensive study by Katich et al. (2023) showed that the mixing states of pyroCb BC remains relatively unchanged over a period of months in the UTLS⁵. Photochemical aging and coating processes that are often the result of secondary aerosol formation by the organic gases associated with fire emissions are not dominant in the stratosphere⁵. Hence, the major finding from this study, that is, E_{abs} approaching 2, is expected to hold valid for both in-cloud and injected pyroCb BC in the stratosphere.”

References:

- Katich, J. et al. Pyrocumulonimbus affect average stratospheric aerosol composition. *Science* 379, 815–820 (2023).
- Sedlacek III, A. J. et al. Using the black carbon particle mixing state to characterize the lifecycle of biomass burning aerosols. *Environ. Sci. Technol.* **56**, 14315–14325 (2022).

2. In my opinion, the paper is often missing the “why.” For example, when describing the previous studies of BC aerosol coatings, a sentence or two to describe how these coatings arise and what they are made of would be appropriate. When describing the observed differences between pyroCb aerosols and those from other wildfires and other BC sources, providing a physical mechanism that would lead to these differences would create a more satisfying discussion. Without describing, hypothesizing, or reiterating the physical mechanisms that lead to these differences, readers may be tempted to chalk up these differences to the small sample size.

We agree that more discussion on the physical mechanisms would improve the context of this paper. Taking this suggestion into consideration, in the revised manuscript, we have added the following paragraph (see lines 163 – 175) regarding the mechanism behind the thick coatings of pyroCb BC and why they differ from non-pyroCb and urban BC emissions:

“Our findings are consistent with Katich et al.’s analysis of pyroCb events over the past 13 years⁵. Their findings unraveled two distinctive features of pyroCb BC: first, they possess extremely thick coatings that differ from non-pyroCb wildfire-sourced BC, and second, the mass concentration ratio of BC to organics in a pyroCb plume—a critical parameter for the modeling of plume radiative impacts—remains a constant at $\sim 0.016 \pm 0.008$. This constant ratio has been observed to sustain over a wide range of altitude, temperature, and plume age, implying that secondary processes of organic aerosol formation is not a dominant coating mechanism for BC in pyroCb plumes post-injection to the UTLS. Informed by these observations, the current consensus holds that coagulation and condensation of low-volatility gases within the strong convective cell of a pyroCb drives the external coating mechanism of BC to a near-stable $R_{BC,i}$. In contrast, BC emissions from urban sources and non-pyroCb wildfires have been observed to undergo large changes in their BC to organics mass ratio both in plume and post-emission because of secondary organic aerosol formation^{24,26,27}.”

We have also added discussion on how the mixing state of the pyroCb investigated in this study compares with other pyroCb events and hypothesize that our results will also be applicable to other pyroCb cases. Please see lines (265 – 274) for this added context.

“A limitation of this study is its focus on a singular fresh pyroCb event and its comparison with a limited dataset of urban and wildfire events. The freshness of the pyroCb plume investigated here could raise a question regarding the BC particles in the plume being at or near their maximal coating thickness, and that further aging of the plume could result in a loss of coating mass and decreased E_{abs} ¹³. However, the comprehensive study by Katich et al. (2023) showed that the mixing states of pyroCb BC remains relatively unchanged over a period of months in the UTLS⁵. Photochemical aging and coating processes that are often the result of secondary aerosol formation by the organic gases associated with fire emissions are not dominant in the stratosphere⁵. Hence, the major finding from this study, that is, E_{abs} approaching 2, is expected to hold valid for both in-cloud and injected pyroCb BC in the stratosphere.”

References:

- Katich, J. et al. Pyrocumulonimbus affect average stratospheric aerosol composition. *Science* 379, 815–820 (2023).
- Ditas, J. et al. Strong impact of wildfires on the abundance and aging of black carbon in the lowermost stratosphere. *Proc. Natl. Acad. Sci.* **115**, E11595–E11603 (2018).
- Lee, A. K. et al. Formation of secondary organic aerosol coating on black carbon particles near vehicular emissions. *Atmospheric Chem. Phys.* **17**, 15055–15067 (2017).

Cheng, Y. *et al.* Size-resolved measurement of the mixing state of soot in the megacity Beijing, China: diurnal cycle, aging and parameterization. *Atmospheric Chem. Phys.* **12**, 4477–4491 (2012).

Sedlacek III, A. J. *et al.* Using the black carbon particle mixing state to characterize the lifecycle of biomass burning aerosols. *Environ. Sci. Technol.* **56**, 14315–14325 (2022).

Hodshire, A. L., Akherati, A., Alvarado, M. J., Brown-Steiner, B., Jathar, S. H., Jimenez, J. L., Kreidenweis, S. M., Lonsdale, C. R., Onasch, T. B., Ortega, A. M., and Pierce., J. R. “Aging Effects on Biomass Burning Aerosol Mass and Composition: A Critical Review of Field and Laboratory Studies”, *Environmental Science & Technology*. 2019 53 (17), 10007-10022. DOI: 10.1021/acs.est.9b02588

Minor/Editorial Points:

1. L40: Consider rewording the sentence to make clear that the convection is related to the heat generated by the wildfire, not heat in general.

We have changed “which form from heat driven convection” to “which form from heat driven convection stemming from wildfires”.

2. L52: What types of previous studies showed this? Modeling, in situ, etc.?

These studies are *in-situ*, we have added this for clarity.

References:

Hodshire, A. L., Akherati, A., Alvarado, M. J., Brown-Steiner, B., Jathar, S. H., Jimenez, J. L., Kreidenweis, S. M., Lonsdale, C. R., Onasch, T. B., Ortega, A. M., and Pierce., J. R. “Aging Effects on Biomass Burning Aerosol Mass and Composition: A Critical Review of Field and Laboratory Studies”, *Environmental Science & Technology*. 2019 53 (17), 10007-10022. DOI: 10.1021/acs.est.9b02588

Christian, K., Yorks, J., and Das. S. “Differences in the Evolution of Pyrocumulonimbus and Volcanic Stratospheric Plumes as Observed by CATS and CALIOP Space-Based Lidars” *Atmosphere* 2020 11, 1035; doi:10.3390/atmos11101035

These references have been added.

REVIEWERS' COMMENTS

Reviewer #1 (Remarks to the Author):

The revisions improved the clarity and impact of the article.

Reviewer #2 (Remarks to the Author):

The authors have appropriately incorporated my suggestions into the revised manuscript. I don't have any additional comments.

Reviewer #3 (Remarks to the Author):

The points raised in my initial review have been sufficiently addressed by the authors. I recommend publication. Nice work!